# Graph-based Subset Selection for Efficient Training of Gene Perturbation Models

## Abstract

Genomic studies face a vast hypothesis space, while interventions such as gene perturbations remain costly and time-consuming. To accelerate such experiments, gene perturbation models predict the transcriptional outcome of interventions. Since constructing the training set is challenging, active learning is often employed in a "lab-in-the-loop" process. While this strategy makes training more targeted, it is substantially slower, as it fails to exploit the inherent parallelizability of Perturb-seq experiments. Here, we focus on graph neural network–based gene perturbation models and propose a subset selection method that, unlike active learning, selects the training perturbations in one shot. Our method chooses interventions that maximize an architecture-aligned reachability proxy over the input knowledge graph, motivated by how supervision can propagate through graph-based perturbation predictors such as GEARS. The selection criterion is defined over the input knowledge graph and is optimized with submodular maximization, ensuring a near-optimal guarantee. Across four retrospective Perturb-seq datasets, GRAPHREACH provides a one-shot alternative to multi-round active learning, corresponding to a reduction of several months under typical Perturb-seq timelines, while producing more stable perturbation selections and maintaining competitive predictive accuracy under the fixed-budget setting studied here.

## 1 Introduction

Genomic research enables the study of genetic factors underlying various health conditions, opening new avenues for therapeutic development. Techniques such as CRISPR interference and PerturbSeq (Barrangou & Doudna, 2016) have revolutionized the landscape of genomic experimentation by enabling high-throughput screenings. However, the majority of CRISPR-based PerturbSeq experiments are restricted to hundreds of single gene perturbations (Dixit et al., 2016; Adamson et al., 2016; Peidli et al., 2022) due to budget and time constraints, despite having over 20,000 potential gene targets. To assist exploring this vast space, machine learning models have been developed to predict the outcome of gene perturbations on a given cell (Bereket & Karaletsos, 2024; Gaudelet et al., 2024; Lopez et al., 2023; Lotfollahi et al., 2023).

Recent methods often leverage graph-based models that use gene-gene interaction networks for perturbation prediction (Roohani et al., 2024). They require a substantial number of training samples that stem from genomic experiments pertaining to a specific hypothesis, rendering training costly and time-consuming because it requires feedback from a wet-lab. Active learning has been proposed to address this problem (Huang et al., 2024) by defining an iterative interaction between the wet lab and the base model to run experiments for genes that will optimize the training, as shown in Fig. 1a. This ensures that resources are not wasted in non-informative genomic experiments. However, in this setting, the training can take months, because despite the process having typically a few iterations ($\leq 5$) (Huang et al., 2024), each iteration translates to a CRISPRi-based PerturbSeq experiment that may take 3-5 weeks (Gasperini et al., 2019; Huang et al., 2024), without accounting for operational delays due to communication or computational hurdles like model retraining.

Another drawback of active learning is that it relies on a randomly initialized model to decide the first batch of genes. As a result, different runs can yield substantially different training sets, which undermines repro-

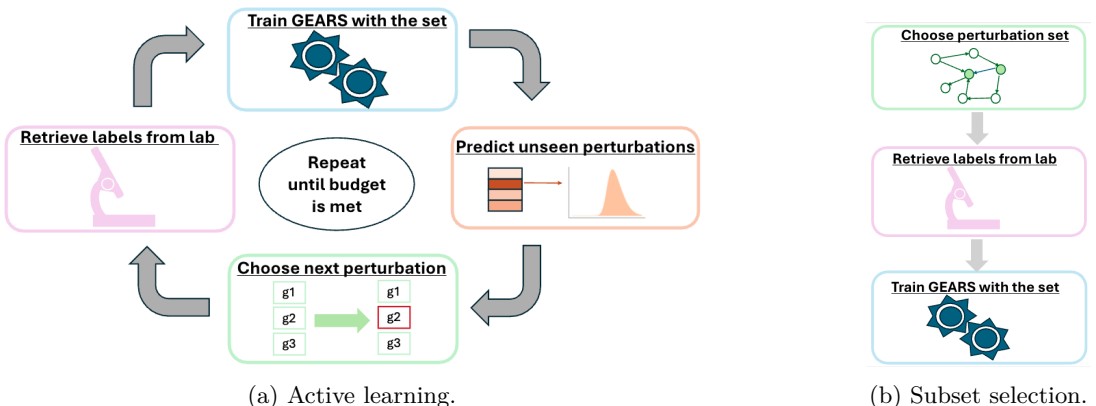

(a) Active learning.

(b) Subset selection.

Figure 1: The difference between active learning and subset selection for training GEARS.

ducibility and limits the reusability of collected data in downstream analyses. To mitigate this instability, we need selection strategies that decouple the model's output from the training set choice. Density-based active learning is model-independent (Hacohen et al., 2022), but it does not apply here because genes lack predefined features; their embeddings are learned by the model itself, so active learning still operates in a model-dependent space (Huang et al., 2024).

In response, we propose a subset selection strategy that builds the training set upfront, without model input. This approach offers several practical benefits:

- **Less experimental time.** Because Perturb-seq assays are parallelizable, selecting all perturbations at once allows the experiments to conclude in approximately the time of a single active-learning cycle, corresponding to a months-level reduction under typical Perturb-seq timelines in the retrospective acquisition setting studied here.

- **Reduced operational complexity.** It avoids repeated cycles of coordination between the wet-lab and the model, model retraining, and acquisition computations, reducing both logistical overhead and potential errors.

- **Greater stability.** By decoupling the selection strategy from model initialization it ensures consistent perturbation sets across runs, making the resulting experimental design easier to justify and adapt to related experiments with overlapping perturbation spaces.

In this work, we use the gene knowledge graph as the selection space because it is already part of the input used by graph-based perturbation predictors such as GEARS (Roohani et al., 2024). We therefore formulate training-set construction as a one-shot graph subset-selection problem. Before observing the perturbation outcomes, we choose a fixed-budget set of candidate interventions using only the candidate list and the knowledge graph as seen in Fig. 1b. The proposed model, GRAPHREACH , selects the gene perturbations whose receptive fields cover the largest number of nodes in the knowledge graph. This objective is motivated by the graph-propagation module, where supervision on selected perturbations can affect embeddings within their receptive fields. Thus, reachability is used as an architecture-aligned proxy for potential supervision propagation. The resulting objective is monotone and submodular, allowing greedy optimization with the standard near-optimality guarantee for submodular maximization.

Overall, we address the problem of training-set selection for graph-based perturbation predictors under a fixed experimental budget to improve accuracy. Our contributions are as follows:

- A one-shot graph-based subset-selection method for training graph-based gene perturbation predictors under a fixed Perturb-seq budget.

- A retrospective empirical comparison on four Perturb-seq datasets using GEARS as the base model, with two active learning methods (one of which is state of the art in our problem).

The results indicate that GRAPHREACH reduces the number of acquisition cycles to one, corresponding to a months-level reduction under typical Perturb-seq timelines, while producing substantially more stable perturbation panels and maintaining competitive predictive accuracy in the evaluated fixed-budget setting.

## 2 Related Work

The body of related work can be broadly divided into two categories. The first concerns optimal experimental design, where the objective is to choose a set of perturbations from a large action space in order to maximize an expected outcome variable, such as T-cell activation (Mehrjou et al., 2021). Methods such as Bayesian optimization and online learning have been applied in this context (Pacchiano et al., 2022; Lyle et al., 2023; Pacchiano et al., 2022), and are evaluated based on the number of high-reward interventions discovered. These algorithms are not applicable in our case because they require a plethora of experiment rounds (e.g., 40), which are feasible with CRISPRi experiments but not when it is combined with PerturbSeq. Moreover, they focus on the experimental success of the retrieved selection rather than the efficacy of the final learning model, which typically predicts scalar values such as phenotypes not multidimensional vectors as in our problem.

The second branch of related literature focuses on efficient training strategies for models that predict the gene expression profile following perturbation in single cells. In this setting, the aim is not to optimize a causal phenotype but to train a model that generalizes well to unseen perturbations. This is particularly useful in Perturb-seq experiments, which measure the transcriptomic effects of perturbations via single-cell RNA sequencing. Due to the high cost and long duration of each experimental cycle, active learning methods have been proposed to efficiently select the most informative perturbations for training. The closest method to our approach is ITERPERT (Huang et al., 2024), which uses active learning and prior multimodal knowledge to build a train set for GEARS. Since each iteration can take 3–5 weeks in the wet lab, the number of iterations is reduced to 5, in contrast to around 50 in optimal experimental design settings (Mehrjou et al., 2021). The problem is addressed from the perspective of active learning under budget (Hacohen et al., 2022) with the inclusion of prior imaging and Perturb-seq studies. In fact, prior multimodal data was so effective that it produced state-of-the-art results without active learning, i.e., the model ITERPERT-prior-only.

Graph-based active learning and graph data-selection methods also select nodes for labeling or training in graph-structured datasets. Examples include influence- or coverage-based methods such as GRAIN (Zhang et al., 2021b) and RIM (Zhang et al., 2021a), which connect GNN data selection with message passing, influence maximization, or label propagation. These methods are methodologically related to our use of graph reachability, but they target general graph learning problems such as node classification or label-efficient GNN training, where labels are obtained for graph nodes. Our setting is different: the selected nodes correspond to gene perturbations that must be assayed through Perturb-seq, the downstream target is a high-dimensional post-perturbation expression profile rather than a node label, and the main practical bottleneck is the number of wet-lab acquisition cycles. To the best of our knowledge, these graph active-learning methods have not been evaluated for one-shot construction of training perturbation panels for graph-based gene perturbation predictors.

Our method differs overall from prior work in several ways. First, we propose a method that selects the perturbations prior to model training, thereby requiring only a single experimental round to obtain the labels. Since Perturb-seq experiments can be parallelized, this reduces the typical duration of training the model from 5 months to approximately one month. Second, unlike ITERPERT's strongest reported setting, which uses curated multimodal priors, our approach relies only on the standardized gene knowledge graph already used by the underlying graph-based perturbation model. Thus, we rely solely on the graph that is integral to the model architecture and broadly available, instead of requiring additional dataset-specific modalities such as imaging data. This distinction is important because such multimodal priors may not be available for many new experimental settings, whereas the ontology-based graph is standardized, reusable, and already part of GEARS. Third, by removing model-driven selection, our method achieves stable and reproducible gene selection. This improves the reusability and interpretability of the resulting data. Finally, the proposed method achieves competitive accuracy compared to the state-of-the-art active learning method that does not use prior multimodal data, TYPICLUST (Huang et al., 2024).

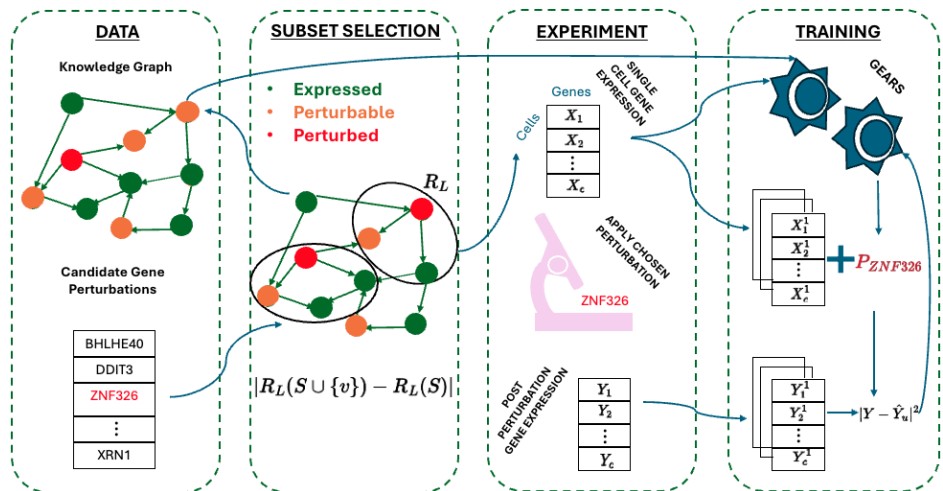

Figure 2: An overview of the subset selection methodology. Our sole input is the knowledge graph from GO51 (Roohani et al., 2024) and a list of candidate perturbations. The subset selection algorithm, such as GRAPHREACH, selects the set of gene perturbations, and they are given to the wet lab for the experimental part. Finally, the single-cell gene expression data is given to GEARS for training and validation.

## 3 Methodology

In this section, we formulate the problem, clarify the theoretical motivation for our approaches and introduce the method for subset selection. An overview of the overall step-by-step subset selection approach can be seen in Fig. 2.

### 3.1 Formulation

We are given a gene knowledge graph $G = (V, E)$, where $|V| = N$ and $|E| = M$, with adjacency matrix $\mathbf{A}$. Each node represents a gene, and a subset $C \subseteq V$ corresponds to candidate gene perturbations that can be experimentally assayed. Our goal is to select a training set $S \subseteq C$, under a budget constraint $|S| = b$ with $b \ll |C|$, to experimentally perturb in a wet-lab setting.

A Perturb-seq screen produces single-cell expression profiles together with perturbation labels. We denote by $\mathcal{D}_{\text{ctrl}}$ the empirical distribution of control cells, and by $\mathcal{D}_u$ the empirical distribution of cells observed under perturbation $u$. Equivalently, the control population of $c$ cells can be represented by $\mathbf{X}^{\text{ctrl}} \in \mathbb{R}^{c \times g}$, while the cells assigned to perturbation $u$ can be represented by $\mathbf{Y}_u \in \mathbb{R}^{c_u \times g}$, where $g$ is the number of measured genes and $c_u$ is the number of cells observed for perturbation $u$.

For each observed perturbed cell $y \sim \mathcal{D}_u$, the model input $x$ is sampled from the control-cell population $\mathcal{D}_{\text{ctrl}}$. The target $y$ is the observed expression vector of a cell annotated with perturbation $u$. Thus, the tuple $(x, u, y)$ is a training example that resembles the supervised signal of the perturbation effect. However, it should be noted that $x$ is not the measured pre-perturbation state of the same cell as $y$ because each measured cell is destroyed. Given the sampled control expression vector $x$ and perturbation $u$, the model predicts a post-perturbation expression vector

$$\hat{y} = f_\theta(x, u) = x + \hat{P}_u,$$

where $\hat{P}_u \in \mathbb{R}^g$ is the model-predicted perturbation-induced effect on expression. This formulation serves as a simplifying abstraction to express perturbation effects as shifts from a baseline state, while not implying true paired observations at the cell level.

The model thus learns a perturbation-effect mapping $\mathcal{P} : C \to \mathbb{R}^g$ that predicts perturbation effects. The model parameters $\theta^S$, trained using only perturbations in $S$, belong to a graph-based gene perturbation

model. We denote by $\ell(\hat{y}, y)$ the model-specific loss between a predicted and observed post-perturbation expression profile. For a perturbation $u$, the loss depends on the model, but can be written in general as:

$$\mathcal{L}_u(\theta^S) = \mathbb{E}_{x \sim \mathcal{D}_{\text{ctrl}}, \, y \sim \mathcal{D}_u} \left[ \ell \left( f_{\theta^S}(x, u), y \right) \right]. \tag{1}$$

Our theoretical analysis below only requires that the loss induces gradients with respect to the graph-propagated representations, and is therefore independent of the precise form of $\ell$. The problem we address is to select the most informative subset of perturbations to experimentally assay, such that the model trained on this subset generalizes well to unseen perturbations $U \subseteq C \setminus S$. Formally, we aim to choose

$$S^\star = \arg \min_{S \subseteq C, \; |S|=b} \sum_{u \in U} \mathcal{L}_u(\theta^S). \tag{2}$$

## 3.2 Subset Selection to Maximize Supervision

Since GRAPHREACH does not use model predictions or perturbation labels at selection time, it is closer in spirit to coverage-based data selection than to uncertainty-based active learning (Sener & Savarese, 2018). Such methods are motivated by the idea that, under suitable assumptions, if each selected training sample covers nearby samples in the input space, then choosing the training set to maximize coverage of the dataset improves prediction. In graph learning, related work has connected supervision at a labeled node to information propagation within its receptive field, i.e., its $k$-hop neighborhood (Wang & Leskovec, 2020; Zhang et al., 2022). Prior work has also shown that the graph distance between training and test nodes is related to GNN generalization (Ma et al., 2021). These findings suggest that selecting perturbations whose receptive fields cover a larger part of the gene knowledge graph may provide a useful proxy for the amount of graph representation space exposed to supervision. In our setting, we therefore use reachability as a model-output-free selection criterion aligned with the graph-propagation mechanism.

In the case of GEARS, let $H^{(r)}$ represent the gene embeddings at graph-propagation layer $r$ of the simplified graph convolution (SGC) used to predict the perturbation $\hat{P}$. Assuming a single SGC layer for clarity (while omitting subsequent layers between $\mathbf{H}$ and the loss $\mathcal{L}$), we have:

$$\mathbf{E} = \mathbf{O}\mathbf{W}_0, \tag{3}$$

$$\hat{\mathbf{A}} = \left( \mathbf{D}^{-1/2}(\mathbf{A} + \mathbf{I}d)\mathbf{D}^{-1/2} \right)^k, \tag{4}$$

$$\mathbf{H} = \hat{\mathbf{A}}\mathbf{E}\mathbf{W}_1, \tag{5}$$

where $\mathbf{D}$ is the diagonal degree matrix, $\mathbf{W}_0 \in \mathbb{R}^{N \times d}$ is the embedding lookup table, $\hat{\mathbf{A}}$ is the normalized adjacency with self-loops to the power of $k = 1$ (which is the default parameter chosen in the GEARS model), $\mathbf{W}_1 \in \mathbb{R}^{d \times d'}$ are the SGC's weights, and $\mathbf{O} \in \mathbb{R}^{N \times N}$ is a row-wise one-hot encoding of the candidate gene perturbations. We base our methodology on the fact that message-passing GNNs update only the representations in $\mathbf{W}_0$ of nodes that are within the receptive field of the supervised nodes. To formalize this, we now state a proposition in which we explicitly calculate the gradient with respect to a particular node representation. The argument is independent of the specific form of $\ell$, and only uses the gradients induced by the resulting objective $\mathcal{L}$ at the graph-propagated representations.

**Proposition 1.** *For any differentiable loss $\mathcal{L}$ applied after the model defined in Equations 3– 5, the gradient with respect to an individual gene embedding $W_0[i]$ is*

$$\frac{\partial \mathcal{L}}{\partial W_0[i]} = \sum_{j=1}^{N} \hat{\mathbf{A}}[j, i] \left( \frac{\partial \mathcal{L}}{\partial H[j]} W_1^\top \right).$$

*Therefore, the gradient of $W_0[i]$ depends only on the gradients of its neighbors in $\hat{\mathbf{A}}$.*

The proof of Proposition 1 can be found in Appendix A. Proposition 1 shows that, under the simplified graph-propagation module, the gradient received by an identity embedding $\mathbf{W}_0[i]$ depends on gradients from

nodes connected through the normalized adjacency operator $\hat{\mathbf{A}}$. Since the loss is evaluated only on selected perturbations, supervision can affect identity embeddings only through the receptive fields of the selected training perturbations.

This motivates reachability as a proxy objective for one-shot subset selection: by selecting perturbations whose receptive fields cover more nodes, we increase the number of identity embeddings that can in principle receive supervision through graph propagation. This argument is specific to the graph-propagation component analyzed above and should not be interpreted as proving that larger reachability necessarily causes lower test error. Rather, it motivates a model-output-free selection criterion aligned with GNN architectures such as GEARS, which are inherently semi-supervised. Therefore, to maximize the reach of the supervision signal, we aim to maximize the number of nodes reached by the training set $S$.

### 3.3 GraphReach

We define our subset selection criterion as a function that maximizes the number of nodes reached by the node set $S$, i.e, the receptive field. Specifically for a new node $v$, the criterion is defined as the additional reachability that $v$ adds in the set of train perturbations $S$

$$\alpha(v, S) = |R_L(S \cup \{v\}) - R_L(S)|, \tag{6}$$

where $R_L$ is the set of nodes reachable from set $S$, based on the number of SGC layers $L$ in the definition of GEARS:

$$R_L(S) = \{v \in V \mid \exists u \in S, \ \exists r \in \{1, \ldots, L\} \text{ such that } (\mathbf{A}^r)_{uv} > 0\}. \tag{7}$$

The subscript $L$ is omitted in subsequent discussion as it remains stable throughout all experiments. In our experiments, the reachability radius $L$ is chosen to match the receptive-field radius induced by the graph-propagation component of GEARS, so that the subset-selection objective and the analyzed architecture use the same notion of graph reachability. The criterion function selects greedily the node $v$ maximizing $\alpha(v, S)$ in every iteration. This is fundamentally suboptimal (Baykal et al., 2021; Kirsch et al., 2019) but like many active learning methods (Lyle et al., 2023; Pacchiano et al., 2022; Tigas et al., 2022; Sussex et al., 2021), we will prove our criterion is monotonic and submodular to achieve a guarantee that the result will be at least $1 - 1/e$ close to the optimal (Nemhauser et al., 1978).

**Proposition 2.** *The reachability function $R_L(S)$ as defined in Eq. 7 is monotonic and submodular.*

The reader is referred to the Appendix B for the whole proof. Consider two sets $S_t \subseteq S_{t+1} \subseteq V$ and any node $v \notin S_{t+1}$. Since $S_t \subseteq S_{t+1}$, every node reachable from $S_t$ is also reachable from $S_{t+1}$, meaning $R(S_t) \subseteq R(S_{t+1})$, which proves monotonicity.

For submodularity, we know by definition that adding a node earlier ($\alpha(v, S_t)$) adds at least as many reachable nodes as adding it later ($\alpha(v, S_{t+1})$). This is because nodes that are newly reached by $v$ when it is added to $S_t$ may already belong to the additional region covered by $S_{t+1}$, i.e. $|R(v) \cap (R(S_{t+1}) \setminus R(S_t))| \geq 0$. This means that any node that appears in $R(v)$ and in $(R(S_{t+1}) \setminus R(S_t))$ was new for $R(S_t)$ but is not new for $R(S_{t+1})$. Hence, the additional reachability that $v$ can bring to $S_{t+1}$ compared to the one it can bring to $S_t$ is diminished. This sketches out the proof that the reachability is submodular.

Despite the theoretical guarantee, exact greedy maximization of reachability over the graph requires recomputing the marginal gain $\alpha(v, S)$ for every remaining candidate perturbation $v \in C \setminus S$ at every selection step. This leads to $\mathcal{O}(b|C|)$ marginal-gain evaluations, where $b$ is the perturbation budget, which can be substantial depending on the graph size and the experiment's scope. To this end, we employ the cost-effective lazy forward strategy for submodular maximization (Leskovec et al., 2007). The method maintains a priority queue of cached marginal gains. Since the reachability function is submodular, the marginal gain of any candidate can only decrease as the selected set $S$ grows. Therefore, a marginal gain computed at an earlier iteration is an upper bound on the candidate's current gain. At each step, the candidate with the largest cached gain is popped from the queue and its true current gain is recomputed. If this recomputed gain is still at least as large as the next largest cached gain in the queue, the candidate is selected; otherwise, its cached

---

**Algorithm 1** GRAPHREACH

---

**Require:** Candidate perturbations $C$, graph $G = (V, E)$, budget $b$, reachability radius $L$
**Ensure:** Selected perturbation set $S$
1:  $S \leftarrow \emptyset$
2:  $R \leftarrow \emptyset$
3:  Initialize an empty max-priority queue $Q$
4:  **for** $v \in C$ **do**
5:      $\Delta[v] \leftarrow |R_L(\{v\})|$
6:      Insert $v$ into $Q$ with priority $\Delta[v]$
7:  **end for**
8:  **while** $|S| < b$ and $Q$ is not empty **do**
9:      $v \leftarrow \text{POPMAX}(Q)$
10:     $\delta \leftarrow |R_L(\{v\}) \setminus R|$
11:     **if** $Q$ is empty **or** $\delta \geq \text{MAXPRIORITY}(Q)$ **then**
12:         $S \leftarrow S \cup \{v\}$
13:         $R \leftarrow R \cup R_L(\{v\})$
14:     **else**
15:         Insert $v$ into $Q$ with priority $\delta$
16:     **end if**
17: **end while**
18: **return** $S$

---

gain is updated and it is inserted back into the queue. Thus, the lazy strategy returns the same selection as exact greedy, while avoiding many redundant marginal-gain computations.

The final subset selection algorithm, called GRAPHREACH from graph reachability, is shown in Alg. 1. It should be noted that the method is easily extended to combinatorial perturbations. In this case, each candidate perturbation corresponds to a set of perturbed genes, and its reachability is defined as the union of the nodes reached by those genes. If two selected candidates reach overlapping nodes, the union ensures that each reached gene is counted only once.

## 4  Experiments

As mentioned in the introduction, subset-selection is significantly faster than active learning due to high-throughput genomics platforms like Perturb-seq being explicitly designed for parallelized experiments. This translates to a many-fold acceleration in our context. Besides speed, here we quantify the changes in other dimensions of the problem, addressing these questions:

- **Accuracy**: How is the generalization of the model affected by the training procedure?

- **Stability**: How much do the proposed genomic experiments change throughout different runs?

- **Robustness**: Is noise in the knowledge graph able to erode the accuracy of the model?

To this end, in this section, we first describe the data used for the experiments, including the gene perturbation datasets and the knowledge graph, the benchmarks used for comparison, as well as the experimental design and the evaluation methods. Afterwards, we showcase the performance of the methods and analyze them to derive conclusions about the soundness of the proposed techniques. The code of the experiments, along with details on computing infrastructure, is provided in the supplementary material.

### 4.1  Data

We test our methods in four single-cell genomic datasets stemming from PerturbSeq experiments, following the literature (Roohani et al., 2024) as seen in Tab. 1. The datasets are diverse in terms of the number of

Table 1: The number of distinct gene perturbations in each datasets (single or combinatorial) along with the average number of cells (samples) per perturbation.

| Dataset | Perturbation Number | Average Cell Count |
|---------|---------------------|--------------------|
| **Adamson** | 81 | 800 |
| **K562** | 1,087 | 150 |
| **RPE1** | 1,535 | 105 |
| **Norman** | 277 | 322 |

perturbations and the number of samples per perturbation. The **Norman** dataset includes combinatorial perturbations (up to 2 genes) and the rest contain data on single perturbations.

Akin to the literature, the graph is based on pathway information from GO51 (Consortium, 2004). Each gene is associated with a number of pathway GO51 terms. The Jaccard similarity between the sets of pathways of two genes, i.e., the fraction of shared pathways, is used to calculate the strength of the edge between them. The graph is sparsified by keeping a predefined number of the most important neighbors for each node. The final graph contains over 9,800 nodes and 200,000 edges, exactly as in (Roohani et al., 2024). It should be noted that GEARS utilizes the whole knowledge graph to learn representations and uses the gene perturbation datasets for supervision only in a semi-supervised manner. GRAPHREACH follows suit and utilizes the whole graph for selection.

## 4.2 Benchmarks

As stated in Section 2, there is currently still a lack of methods addressing our exact setting: one-shot subset selection for training graph-based gene perturbation models under a fixed experimental budget. We do not include the full *ITERPERT* configuration as a benchmark because its main contribution relies on fusing the model kernel with several external multimodal priors, including additional Perturb-seq data, optical pooled screens, scRNA-seq atlases, protein structure, PPI networks, and literature embeddings. These modalities are not uniformly available across datasets and are not part of the standard input required by a gene perturbation model. We therefore restrict all methods to the information available in our setting: the candidate perturbation set, the gene knowledge graph, and, for active-learning baselines, the current model representation. In this setting, Huang et al. (2024) report *TYPICLUST* as the strongest active-learning baseline. Thus, we rely on the following benchmarks:

- BASELINE represents the random selection from the available gene perturbations. This is the prevalent practice because of its speed and simplicity. It represents the vanilla usage of GEARS.

- ACS-FW (Pinsler et al., 2019) is a Bayesian batch active learning model. It selects perturbations such that the new posterior distribution of the model's parameters approximates the expected posterior when the whole dataset is available. We utilized the version from the BMDALreg library (Holzmüller et al., 2023), which was one of the top-performing methods in similar experiments (Huang et al., 2024).

- TYPICLUST (Hacohen et al., 2022) is the state-of-the-art active learning method on this problem as it has outperformed 8 active learning models (Huang et al., 2024). The algorithm clusters the candidate perturbations based on the final graph layer representation from GEARS. Within each cluster, the typicality is quantified as the inverse of the average distance between each sample and an example's $K$-nearest neighbors, with $K = 20$. The most typical sample is selected.

## 4.3 Design

For GRAPHREACH and BASELINE, the train and validation sets are defined before the acquisition of the single-cell data and they require no interaction with the wet-lab. Thus, they are expected to take around

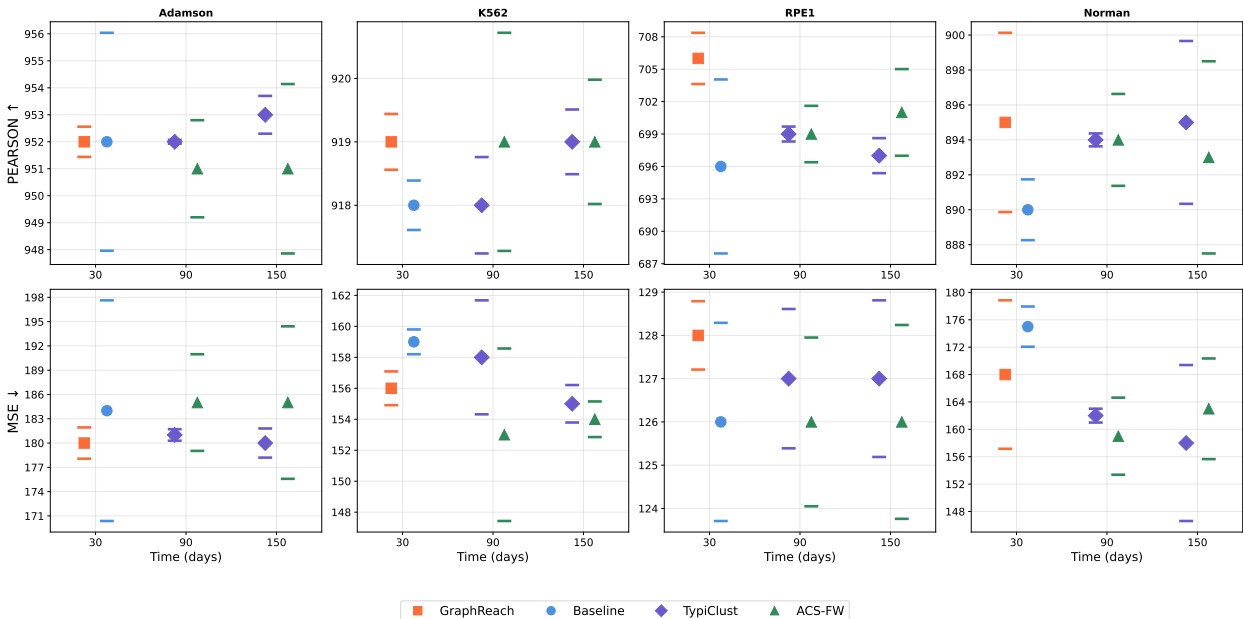

Figure 3: Accuracy ($10^3$ for both metrics) in the test set with confidence intervals compared to the total physical time required for the training due to wet-lab iterations. GRAPHREACH and BASELINE do not require model-input hence they are trained with only one cycle of wet lab experiments taking 30 days. TYPICLUST and ACS-FW inherently need numerous cycles, and they are each run for 90 and 150 days to highlight the role that the number of cycles plays in the accuracy.

30 days, which corresponds to the time the PerturbSeq experiment takes, since running GRAPHREACH at the beginning and training GEARS at the end takes negligible time. For all methods, 25% of the gene perturbations are used for training and 5% for validation, 10% are kept for testing, leaving 60% of the dataset unutilized. This is imperative to perform an effective comparison between the selected train sets. If we increase the train set size, the methods converge unavoidably to very similar final gene selections because the available choices are limited. Hence, we leave sufficient space between choices to highlight the differences between the strategies. We use a 10-fold cross validation to adjust the 10% test set and take the average performance. The whole experiment is repeated for 3 different random seed initializations (including model initialization and data splits).

For the active learning methods, in accordance to (Huang et al., 2024), we keep the initial 10% test set constant throughout the cycles to ensure there is no interaction between the strategies and the evaluation. We train the model a number of times sequentially, each time adding to the train and validation set a percentage of the available perturbations until they reach 25% and 5%, respectively. In order to quantify the relationship between the training cycles and final model accuracy, active learning methods are run for 3 and 5 cycles, the latter being the main choice in the literature. The models with 5 cycles receive 5% of the perturbations in each cycle for training and 1% for validation, while for 3 cycles the numbers are 8.3% and 1.7% respectively. Models trained on 5 cycles are expected to take 150 days, and 3 cycles correspond to 90 days. The methods are evaluated based on the performance of GEARS trained on the final cycle. It should be noted that no active learning method can go less than 60 days, as 1 training cycle (30 days) means no input from the model.

Following the literature, the evaluation metrics for performance are Pearson correlation and MSE, which is computed on the top 20 most differentially expressed genes, i.e., the genes that exhibit the biggest expression differences in the experiment. This is a common practice since the vast majority of the genes have zero expression, meaning the results would be skewed and the differences minuscule had we computed them for more genes (Roohani et al., 2024). We evaluate stability by quantifying how much the chosen gene perturbations diverge between different seed initializations for each method. To this end, we quantify the

consistency of each method by measuring the overlap between the resulting gene sets using the average pairwise Jaccard similarity defined in the appendix. To evaluate robustness to noise, we performed the same experiments where GRAPHREACH runs on corrupted versions of the knowledge graph. Specifically, we delete 5% and 10% of the edges and add the same amount of fake ones, to represent a common setting of errors in the graph. In all experiments, we use the default GEARS parameters and implementation[1].

### 4.4 Results

#### 4.4.1 Accuracy

The overall tradeoff between accuracy and efficiency is visualized in Fig. 3. For Pearson correlation, we observe that GRAPHREACH achieves competitive performance, performing on par with or better than the benchmarks in three of the four datasets, despite taking a fraction of the time. Compared with the BASELINE, the only benchmark with similar efficiency, GRAPHREACH improves or matches performance in most dataset–metric evaluations, with the detailed raw Pearson and MSE values reported in Appendix F. The baseline also tends to exhibit larger confidence intervals, indicating greater variability between runs.

On average, the active learning methods perform on par with each other, with TYPICLUST being better in **Adamson** and **Norman** and ACS-FW being better in the rest. Another observation for ACS-FW is that it remains constant for 3 and 5 cycles for **Adamson**, 3 cycles are better than 5 in **Norman** while the results are split in **K562** and **RPE1**. We believe that this happens because training on 3-cycles increases the amount of samples the model sees per cycle and this allows GEARS to provide better uncertainty estimates which are integral for ACS-FW. In contrast, TYPICLUST tends to deteriorate as we constrain the days with only one exception in **RPE1** for Pearson correlation. That said, ACS-FW has considerably larger confidence intervals (competing with the BASELINE) throughout all experiments, possibly due to the sensitivity to GEARS' uncertainty estimation. We analyze further this instability in the coming sections. We thus deem TYPICLUST the second best method, as it is considerably more stable and achieves competitive performance.

Note that, as mentioned above, active learning requires by default a number of cycles to run and hence it can not achieve similar efficiency to GRAPHREACH and we can not compare them head-to-head. However, if we constrain the experiments to up to 90 days, GRAPHREACH performs better than active learning in 5 out of 8 evaluations. Given that on average active learning deteriorates as we constrain the days, we can deduce that GRAPHREACH showcases the best tradeoff between accuracy and efficiency.

#### 4.4.2 Stability

The average Jaccard similarity between the perturbation sets selected through all cycles can be found in Tab. 2. The random seed affects the model and the k-folds of the data, hence the set of genes retrieved by GRAPHREACH differ solely due to different splits. In contrast, active learning methodologies exhibit variability across different runs that approximate the BASELINE, which is a random selection. This stability has practical value in the experimental-design setting. The selected perturbation set determines which genes will be assayed in the wet lab and therefore which costly training data will be generated. In this sense, the perturbation panel is itself an output of the method. If a model-based active-learning strategy produces substantially different panels under different random initializations, then the recommended experiment depends strongly on stochastic model artifacts rather than on a robust signal in the candidate perturbation space. Given the cost and duration of Perturb-seq experiments, this instability risks allocating wet-lab resources to experiments that are sensitive to arbitrary initialization choices. A concrete downstream scenario is the design of a training set for graph-based perturbation prediction. With GRAPHREACH, the selected perturbations are tied to the structure of the candidate perturbation space and to the graph neighborhoods used by the predictor. Therefore, the resulting panel can provide a documented set for related studies with overlapping perturbation spaces, for example when extending to nearby pathways, related cell types or states, or other graph-based perturbation models. In contrast, if the selected panel changes substantially across

---

[1]https://github.com/snap-stanford/GEARS/tree/master

Table 2: Average Jaccard similarity on gene selections through different random seed initializations.

| Data | Base line | ACS FW | Typi Clust | Graph Reach |
|---|---|---|---|---|
| **Adamson** | 0.15 | 0.15 | 0.15 | **0.75** |
| **K562** | 0.10 | 0.13 | 0.10 | **0.78** |
| **RPE1** | 0.10 | 0.13 | 0.10 | **0.76** |
| **Norman** | 0.11 | 0.13 | 0.15 | **0.95** |

Table 3: Performance under different noise levels in the knowledge graph, as represented by random removal and addition of edges. Results are multiplied by $10^3$ for readability.

| Dataset | Noise (%) | Pearson ($\uparrow$) | MSE ($\downarrow$) |
|---|---|---|---|
| | 0 | $952 \pm 1$ | $180 \pm 2$ |
| **Adamson** | 5 | $951 \pm 1$ | $181 \pm 6$ |
| | 10 | $947 \pm 1$ | $196 \pm 8$ |
| | 0 | $919 \pm 1$ | $156 \pm 1$ |
| **K562** | 5 | $919 \pm 2$ | $154 \pm 1$ |
| | 10 | $920 \pm 2$ | $153 \pm 2$ |
| | 0 | $706 \pm 3$ | $128 \pm 1$ |
| **RPE1** | 5 | $701 \pm 4$ | $129 \pm 1$ |
| | 10 | $704 \pm 6$ | $126 \pm 2$ |
| | 0 | $895 \pm 5$ | $168 \pm 10$ |
| **Norman** | 5 | $893 \pm 13$ | $170 \pm 18$ |
| | 10 | $894 \pm 13$ | $163 \pm 13$ |

random initializations, it is less clear whether it can be used as a starting point for related experimental designs.

### 4.4.3 Robustness

The results from the robustness experiments are in Tab. 3. We observe that while GRAPHREACH under-performs in noisy settings in the **Adamson** dataset, it actually stays close to or even surpasses the original in the rest. We hypothesize that this happens because random edges may have a relatively high probability of connecting previously distant parts of the graph and may therefore further increase the coverage that GRAPHREACH is able to achieve. So, we observe satisfactory empirical robustness of GRAPHREACH. Similar effects of random perturbations on approximation quality have been observed in studies of vertex cover and related graph problems (Shi et al., 2018).

## 5 Conclusion

We studied one-shot training-set construction for graph-based gene perturbation predictors under fixed Perturb-seq budgets. GRAPHREACH uses only the candidate perturbation list and the gene knowledge graph to select perturbations whose receptive fields maximize a reachability objective. In retrospective experiments with GEARS across four Perturb-seq datasets, GRAPHREACH reduced the number of required acquisition cycles compared with multi-round active learning, produced substantially more stable perturbation panels, and achieved predictive accuracy comparable to random selection and active-learning baselines in the evaluated setting. These results support GRAPHREACH as a simple graph-only alternative for GEARS-style perturbation modeling when wet-lab acquisition cycles dominate the experimental cost.

For future directions, the method could also extend beyond the current GEARS setting. Since GRAPHREACH only requires a candidate perturbation set and a graph defining relationships between perturbations, it can in principle be applied with other biological knowledge graphs, such as pathway, regulatory, co-expression, or protein-interaction graphs, provided that the graph is relevant to the perturbation predictor. This makes

graph-based subset selection a possible tool for perturbation settings where external biological structure is available but repeated wet-lab acquisition cycles are impractical.

Another potential future direction is for single-cell foundation models. If the foundation model is paired with an external biological graph, the same reachability-based formulation could be used to select perturbations for fine-tuning or evaluation. If no graph is available, one possible extension is to construct a similarity graph from pretrained model embeddings before fine-tuning, and then apply graph-based subset selection on this induced graph. This would make the selection model-dependent, so it should be studied separately. Training a foundation model from scratch is a different problem, because the relevant representation space may not exist before pretraining.

Finally, our evaluation is retrospective and model-specific. Prospective wet-lab validation, evaluation with additional graph-based perturbation predictors, and systematic study of alternative knowledge graphs remain important directions for future work.

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

## Appendix

## A    Proof of Proposition 1

We begin by considering the gradient of the loss $\mathcal{L}$ with respect to the identity embeddings $\mathbf{W}_0$ as a function of $\frac{\partial \mathcal{L}}{\partial \mathbf{H}}$:

$$\frac{\partial \mathcal{L}}{\partial \mathbf{W}_0} = \mathbf{O}^\top \left( \frac{\partial \mathcal{L}}{\partial \mathbf{E}} \right), \tag{8}$$

$$\frac{\partial \mathcal{L}}{\partial \mathbf{E}} = \hat{\mathbf{A}}^\top \left( \frac{\partial \mathcal{L}}{\partial \mathbf{H}} \mathbf{W}_1^\top \right). \tag{9}$$

Thus, the full gradient becomes:

$$\frac{\partial \mathcal{L}}{\partial \mathbf{W}_0} = \mathbf{O}^\top \hat{\mathbf{A}}^\top \frac{\partial \mathcal{L}}{\partial \mathbf{H}} \mathbf{W}_1^\top. \tag{10}$$

For an individual embedding row $\mathbf{W}_0[i]$ and since $\mathbf{O}$ is one-hot, this expands to:

$$\frac{\partial \mathcal{L}}{\partial \mathbf{W}_0[i]} = \sum_{j=1}^{N} \hat{\mathbf{A}}[j,i] \cdot \left( \frac{\partial \mathcal{L}}{\partial \mathbf{H}[j]} \mathbf{W}_1^\top \right), \tag{11}$$

which shows that the gradient of $\mathbf{W}_0[i]$ depends only on the gradients of its neighbors in $\hat{\mathbf{A}}$. Since the loss is evaluated only on the selected perturbations, nonzero output gradients are induced only by supervised perturbation nodes:

$$\frac{\partial \mathcal{L}}{\partial H[j]} = 0 \quad \text{if } j \notin S.$$

Therefore identity embeddings can receive gradient only through graph propagation from the selected training perturbations.

## B    Proof of Proposition 2

Following the definitions of the main paper, let $R$ be the set function of reachability in the graph, and we have subsets of nodes $S_t \subseteq S_{t+1}$. We can start from the definition of submodularity:

$$R(S_t \cup \{u\}) \setminus R(S_t) \supseteq R(S_{t+1} \cup \{u\}) \setminus R(S_{t+1}) \tag{12}$$

and continue by reformulating the right and left-hand side of Eq. 12 as follows,

$$R(S_t \cup \{u\}) \setminus R(S_t) = \big( R(S_t) \cup R(\{u\}) \big) \setminus R(S_t)$$
$$= R(\{u\}) \setminus R(S_t),$$
$$R(S_{t+1} \cup \{u\}) \setminus R(S_{t+1}) = \big( R(S_{t+1}) \cup R(\{u\}) \big) \setminus R(S_{t+1})$$
$$= R(\{u\}) \setminus R(S_{t+1}).$$

Plugging these reformulations back into Eq. 12 produces,

$$R(\{u\}) \setminus R(S_t) \supseteq R(\{u\}) \setminus R(S_{t+1}). \tag{13}$$

We shall now derive the reformulated definition of submodularity in Eq. 13 to show that reachability is submodular.

We start with the fact that by definition we have $S_t \subseteq S_{t+1}$ and graph reachability $R$ is monotone, consequently:

$$R(S_t) \subseteq R(S_{t+1}). \tag{14}$$

Now recall the anti-monotonicity property of set difference (as illustrated below for sets $X, Y$, and $Z$):

$$X \subseteq Y \quad \implies \quad Z \setminus X \supseteq Z \setminus Y. \tag{15}$$

So if we add a set difference on Eq. 14 with $R(\{u\})$, we get the desired:

$$R(\{u\}) \setminus R(S_t) \supseteq R(\{u\}) \setminus R(S_{t+1}). \tag{16}$$

Thus,

$$R(S_t \cup \{u\}) \setminus R(S_t) \supseteq R(S_{t+1} \cup \{u\}) \setminus R(S_{t+1}), \tag{17}$$

which proves that reachability is submodular.

## C   Computational Time

The computational time is negligible in our setting, because each sample selection takes less than a minute. That said, *GRAPHREACH* is around 500 times faster than *TYPICLUST*, meaning it scales significantly better with the number of available perturbations. This is important for experiments without predefined candidate perturbations, where the search space can increase from hundreds to tens of thousands.

## D   Model initialization bias

Model-based active learning is sensitive to random initialization in early acquisition cycles. In standard active learning settings, this sensitivity may decrease after many acquisition rounds, as the selected sets gradually stabilize. In CRISPRi + Perturb-seq experiments, however, the number of rounds is severely limited because each cycle can require several weeks of wet-lab time. As a result, early model-dependent choices can have a disproportionate effect on the final selected perturbation set.

Figure 4 illustrates this effect by showing the number of unique perturbations selected across different GEARS initializations on the same train-test split. Across seeds, active learning selects substantially more unique genes than the nominal budget, indicating that the selected training set is strongly influenced by initialization. This limits reproducibility and reduces the reusability of the resulting experimental data across independent runs or laboratories.

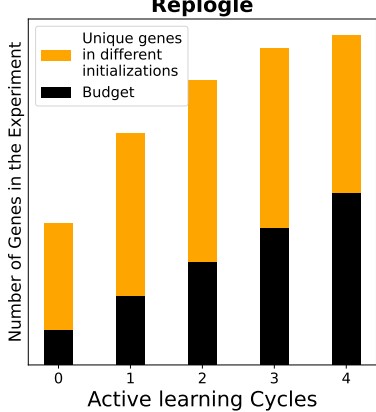
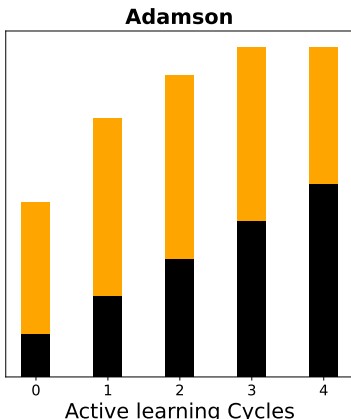

Figure 4: The total number of unique genes perturbed in active learning experiments for 5 different initializations of GEARS, on the same train-test split.

Table 4: Comparison between random selection and GRAPHREACH. Pearson and MSE are reported on the original scale. $\Delta$ values are reported in $\times 10^{-3}$ units; positive values indicate an improvement of GRAPHREACH over random selection.

| Dataset | Random P. | GR P. | $\Delta$P | Random MSE | GR MSE | $\Delta$MSE |
|---------|-----------|-------|-----------|------------|--------|-------------|
| Adamson | $0.952 \pm 0.004$ | $0.952 \pm 0.001$ | 0 | $0.184 \pm 0.014$ | $0.180 \pm 0.002$ | 4 |
| K562 | $0.918 \pm 0.000$ | $0.919 \pm 0.000$ | 1 | $0.159 \pm 0.001$ | $0.156 \pm 0.001$ | 3 |
| RPE1 | $0.696 \pm 0.008$ | $0.706 \pm 0.002$ | 10 | $0.126 \pm 0.002$ | $0.128 \pm 0.001$ | $-2$ |
| Norman | $0.890 \pm 0.002$ | $0.895 \pm 0.005$ | 5 | $0.175 \pm 0.003$ | $0.168 \pm 0.011$ | 7 |
| Average | – | – | 4.0 | – | – | 3.0 |

## E  Infrastructure

The computing infrastructure used for the experiments reported in this paper includes a 13th Gen Intel(R) Core(TM) i9-13900K CPU with 24 cores, two NVIDIA GeForce RTX 4070 with 12GB and a CUDA Version: 12.2, and a RAM of 32 GB on an Ubuntu 22.04.

## F  Detailed Results

Table 4 reports the raw accuracy values for the direct comparison between random selection and GRAPHREACH. Across the four datasets and two metrics, GRAPHREACH improves over random selection in six evaluations, matches it in one, and underperforms in one. The average improvement is $+4 \times 10^{-3}$ in Pearson correlation and $-3 \times 10^{-3}$ in MSE. These differences are consistent with the scale of improvements reported in prior active-learning work for gene perturbation model training. For example, Fig. 4 of Huang et al. (Huang et al., 2024) reports improvements on the order of a few $10^{-3}$ for both MSE and Pearson correlation. Thus, the accuracy differences observed here should be interpreted in the context of the active-learning and subset-selection setting, where the base perturbation predictor is fixed and the intervention is only the choice of training perturbations.

