# OpenReview forum: "Graph-based Subset Selection for Efficient Training of Gene Perturbation Models"
_TMLR — Under review for TMLR_

### Review · Reviewer_Ywyo · 2026-06-08

**Summary Of Contributions:**

The paper studies one-shot subset selection for training graph-based gene perturbation predictors, focusing on GEARS, and proposes GRAPHREACH, a greedy selection rule that maximizes graph reachability of supervised nodes under a budget. The main technical motivation comes from the gradient propagation argument, which links supervision to the receptive field of selected perturbations. Empirically, the paper compares against random selection, ACS-FW, and TypiClust on four Perturb-seq datasets. The strongest aspect is that the paper is simple, easy to deploy, and the experiments are aligned with a realistic wet-lab bottleneck. The biggest issue is that the core methodological contribution and novelty claim appear insufficiently positioned and may substantially overlap with prior work on graph-based one-shot subset selection for perturbation-model training.

**Audience:**

Yes

**Audience Explanation:**

This is clearly in scope for TMLR. Even with my concerns about novelty and positioning, the problem itself is relevant and the experimental findings would be of interest to at least a subset of TMLR readers.

**Claims And Evidence:**

No

**Claims Explanation:**

The main issue is novelty/positioning. The paper presents GRAPHREACH as a new method, but the core contribution, graph-based one-shot subset selection for training genomic perturbation GNNs via a reachability/submodular objective, appears too close to prior work, and this is not adequately discussed. This matters because the paper’s headline claim is not just “this works” but “this is our methodological contribution.” As written, the paper does not make a convincing case that the method is meaningfully distinct.

**Requested Changes:**

1. It is better thet the authors clarify the novelty claim and differentiate this work from prior methods for genomic perturbation model training.
2. Please narrow the paper’s strongest claims to match the evidence. In particular, statements in the Abstract, Introduction, and Conclusion that imply a generally superior approach should be scoped to the specific setting studied in this paper.
3. The authors are encouraged to make the connection between the theory and the actual GEARS architecture more precise.

---

> ### Author Response · Authors · 2026-06-21
> **Answer to reviewer**
>
> Thank you for raising this concern. We agree that the novelty and positioning should be made clearer during the revision.
>
> - 'It is better thet the authors clarify the novelty claim and differentiate this work from prior methods for genomic perturbation model training.'
>
> We revised the related-work discussion to more explicitly distinguish our setting from general graph active-learning methods that use submodularity, such as GRAIN or RIM. However, we are not aware of any other similar method applied in the context of gene perturbation models. If the reviewer has specific prior work in mind, we would be grateful if they could point us to it and will discuss it explicitly.
>
> -  'Please narrow the paper’s strongest claims to match the evidence. In particular, statements in the Abstract, Introduction, and Conclusion that imply a generally superior approach should be scoped to the specific setting studied in this paper.'
>
> We narrowed the strongest claims in the Abstract, Introduction, and Conclusion. The revised manuscript now states that our empirical claims concern the datasets, baselines, and budgeted one-shot Perturb-seq setting studied in the paper, rather than claiming general superiority.
>
> -  'The authors are encouraged to make the connection between the theory and the actual GEARS architecture more precise.'
>
> We clarified the scope of the gradient-propagation argument. The analysis is presented as motivation for reachability as an architecture-aligned proxy for GEARS-style graph propagation, not as a proof that reachability alone determines generalization. We use GEARS as the concrete model because it is the graph-based perturbation predictor evaluated in our experiments. The selection rule itself is model-output-free, meaning it does not utilize any essential part of the GEARS architecture apart from the fact that it s a GNN and the number of message passing layers.

---

> > ### Comment · Reviewer_Ywyo · 2026-06-22
> >
> > Thank you for the detailed responses to my previous concerns. I find your revisions and clarifications satisfactory. The manuscript has been improved through these revisions on my end. I will also consider other reviewers' concerns. Thanks!

---

### Review · Reviewer_44HZ · 2026-06-09

**Summary Of Contributions:**

This paper studies the problem of efficient training-set construction for graph-based gene perturbation models. The authors argue that active learning is not fully suitable for Perturb-seq settings because each acquisition round requires a costly and time-consuming wet-lab cycle. To address this issue, the paper proposes GraphReach, a one-shot subset selection method that chooses gene perturbations before model training. The main idea is to select perturbations that maximize the number of nodes reached in the gene knowledge graph, so that the supervision signal can propagate to more gene embeddings when training a graph-based perturbation predictor such as GEARS. The reachability objective is shown to be monotone and submodular, and the authors optimize it using a lazy greedy algorithm. Experiments on four Perturb-seq datasets suggest that GraphReach can achieve competitive prediction accuracy with substantially fewer wet-lab cycles and more stable perturbation selections than active-learning baselines.

**Audience:**

Yes

**Audience Explanation:**

The paper should be of interest to researchers working on graph neural networks for biological systems, active learning, subset selection, and efficient design of perturbation experiments.

**Broader Impact Concerns:**

There are not concerns that that would require adding a Broader Impact Statement.

**Claims And Evidence:**

No

**Claims Explanation:**

The evidence supports GraphReach as a stable one-shot training-set selector for GEARS, but not the broader claims about wet-lab utility or superiority over simple graph heuristics.

Strength:
 - The paper addresses a practically important problem. In Perturb-seq-based gene perturbation modeling, the main bottleneck is not only computational cost but also the physical time required for wet-lab acquisition. The paper correctly highlights that multi-round active learning can be unattractive in this setting because each round may take several weeks.
 - The proposed one-shot subset selection formulation is well motivated. Since Perturb-seq experiments are highly parallelizable, selecting all perturbations in advance is practically meaningful and can reduce operational complexity compared with iterative model-lab interaction.

Weakness:
 - The novelty of the method is somewhat limited. GraphReach is essentially a maximum coverage objective over L-hop neighborhoods in a graph, optimized by standard submodular greedy maximization. While the application to gene perturbation model training is useful, the algorithmic contribution itself is relatively incremental.
 - The experimental comparison lacks important graph-based baselines, e.g. degree centrality, PageRank, and graph clustering representatives.
 - The real-world acceleration claim is plausible but only indirectly supported: the paper estimates wet-lab time from acquisition cycles, without prospective wet-lab validation showing GraphReach-selected perturbations yield better or comparable biological outcomes under a real budget.
 - Some claims are slightly stronger than the evidence supports. For example, the paper suggests that GraphReach provides months of acceleration without essentially sacrificing accuracy. This is plausible under the simulated acquisition setting, but the actual experimental utility remains only partially validated.

**Requested Changes:**

- Add graph-only baselines, including degree centrality, PageRank, and graph clustering representatives.
 - Improve the figure quality of figure 1 and figure 2.
 - Test GraphReach with another graph-based perturbation model besides GEARS, or soften the claim that the method generalizes broadly to graph-based predictors.
 - Discuss more explicitly that the current evaluation is retrospective and simulated, not a prospective wet-lab validation.

---

> ### Author Response · Authors · 2026-06-21
> **First answer to reviewer**
>
> We thank the reviewer for the helpful comments.
>
> - The novelty of the method is somewhat limited. GraphReach is essentially a maximum coverage objective over L-hop neighborhoods in a graph, optimized by standard submodular greedy maximization. While the application to gene perturbation model training is useful, the algorithmic contribution itself is relatively incremental.
>
> We thank the reviewer for this fair assessment.  We agree that GraphReach is related to existing graph-based active-learning and subset-selection methods, and we cite and discuss these connections more explicitly in the revised manuscript. Our contribution is not to claim a fundamentally new optimization paradigm, but to adapt this paradigm to an important problem: training gene perturbation models under fixed experimental budgets. To the best of our knowledge, this is the first work to study this graph-based one-shot subset-selection formulation in the context of gene perturbation prediction.
>
> - The real-world acceleration claim is plausible but only indirectly supported: the paper estimates wet-lab time from acquisition cycles, without prospective wet-lab validation showing GraphReach-selected perturbations yield better or comparable biological outcomes under a real budget.
> - Some claims are slightly stronger than the evidence supports. For example, the paper suggests that GraphReach provides months of acceleration without essentially sacrificing accuracy. This is plausible under the simulated acquisition setting, but the actual experimental utility remains only partially validated.
> - Discuss more explicitly that the current evaluation is retrospective and simulated, not a prospective wet-lab validation.
>
> We thank the reviewer for this important clarification. We agree that our current evaluation is retrospective and simulated, not a prospective wet-lab validation. The acceleration claim is therefore not meant to imply that GraphReach has already been prospectively tested in a real lab-in-the-loop Perturb-seq campaign. Rather, the claim follows from the experimental structure of the compared strategies: active learning requires multiple sequential acquisition cycles, whereas GraphReach selects the full perturbation panel in one shot. Since Perturb-seq assays can be run in parallel and each acquisition cycle is reported to take several weeks, reducing the number of cycles from multiple rounds to one corresponds to a practical time reduction under the acquisition model used in the paper.
> We revised the Abstract, Introduction, and Conclusion to make this distinction clearer. In particular, we now describe the time reduction as an estimate based on typical Perturb-seq timelines and explicitly state that the evaluation is retrospective and model-specific. We also clarify that prospective wet-lab validation remains necessary to establish the real-world biological utility of GraphReach-selected panels under an actual experimental budget.
>
> - Improve the figure quality of figure 1 and figure 2.
>
> We thank the reviewer for this suggestion. We agree that the clarity and visual quality of Figures 1 and 2 can be improved. In the current revision, we focused on addressing the methodological and experimental concerns, but in the final revised manuscript we will redraw both figures in a cleaner vector format using TikZ.
>
> - Test GraphReach with another graph-based perturbation model besides GEARS, or soften the claim that the method generalizes broadly to graph-based predictors.
>
> We thank the reviewer for this suggestion. We agree that evaluating GraphReach with additional graph-based perturbation predictors would strengthen the empirical evidence. In the current paper, we focus on GEARS because it is a widely used graph-based perturbation predictor and is also the base model used by the closest active-learning work in this setting, which allows a controlled comparison. We therefore revised the manuscript to soften the generality claim: while the selection rule is model-output-free and can in principle be applied to other graph-based perturbation predictors with a suitable knowledge graph and receptive field, our empirical evidence is currently limited to GEARS. We now present evaluation with additional graph-based perturbation models as an important direction for future work.

---

> ### Author Response · Authors · 2026-06-21
> **Second answer to reviewer**
>
> - Add graph-only baselines, including degree centrality, PageRank, and graph clustering representatives.
>
> We thank the reviewer for this suggestion. Here is a comparison with graph-only baselines based on degree centrality, PageRank, and a clustering baseline, where we partition the graph into a number of clusters equal to the budget and select one node from each cluster.
>
> Due to the time constraints of the rebuttal, we ran this additional comparison only with random seed set to 0. To make the comparison fair, we also restrict the reported \textsc{GraphReach} results in this table to seed 0. The same evaluation protocol is otherwise used. We did not include Norman in this additional analysis because of these computational constraints and because Norman is already the least favorable dataset for \textsc{GraphReach} in the main results, so its omission does not remove a favorable case.
>
> | Dataset | Metric | \textsc{GraphReach} | Degree | PageRank | Clustering |
> |---|---:|---:|---:|---:|---:|
> | Adamson | MSE ↓ | **177** | *189* | *189* | 194 |
> | Adamson | Pearson ↑ | **952** | *950* | 948 | 948 |
> | K562 | MSE ↓ | *155* | **154** | 156 | 177 |
> | K562 | Pearson ↑ | *919* | *919* | **933** | 915 |
> | RPE1 | MSE ↓ | **129** | **129** | **129** | 132 |
> | RPE1 | Pearson ↑ | *708* | 702 | 707 | **712** |
> | **Average** | MSE ↓ | **153.7** | *157.3* | 158.0 | 167.7 |
> | **Average** | Pearson ↑ | *859.7* | 857.0 | **862.7** | 858.3 |
> | **Average rank** | -- | **1.75** | *2.33* | 2.50 | 3.42 |
>
> Metrics are reported in (\times 10^{-3}) units. Lower is better for MSE and higher is better for Pearson. Best values are shown in bold and second-best values are italic. Average rank is computed across all dataset--metric pairs, with lower rank better.
> The graph-only baselines are competitive, confirming that the graph structure itself is a useful signal for subset selection. However, \textsc{GraphReach} achieves the best average MSE and the best average rank across the evaluated dataset--metric pairs. PageRank obtains the highest average Pearson, mainly due to its strong result on K562, but it underperforms on most other dataset--metric pairs. Overall, \textsc{GraphReach} provides the strongest performance across metrics.
>
> Regarding the magnitude of the accuracy differences, small numerical gains are expected in this setting because the base perturbation predictor is fixed and the intervention is only the choice of training perturbations. These differences are of the same order as those reported in prior active-learning work for gene perturbation model training. To be specific, Huang et al. (2024) which we rely on for our experimental design and baseline choice, report gains of approximately (-3 \times 10^{-3}) MSE and (+5 \times 10^{-3}) Pearson in their Fig. 4. Thus, the observed differences are consistent with the scale of improvements reported in the relevant active-learning literature for this problem.

---

### Review · Reviewer_UdzV · 2026-06-11

**Summary Of Contributions:**

The paper addresses the training set construction for graph-based gene perturbation models (specifically GEARS).
A competitive approach is active learning ("lab-in-the-loop"), which is slow because each acquisition cycle requires a separate Perturb-seq experiment, and unstable because the 1st iteration is chosen randomly.
The authors introduce GraphReach, a subset selection method that selects all training perturbations in one shot, using only the gene knowledge graph already employed by GEARS.

The method selects perturbations that maximize the number of graph nodes reachable from the training set i.e. the receptive field of the supervision signal. This builds on the intuition that since gradients in the GEARS embedding table propagate only to nodes within the receptive field of supervised nodes, maximizing reachability will maximize the number of non-random embeddings available when predicting unseen perturbations, which should in turn lead to better generalization (i.e. predicting the outcome of an unseen perturbation).

Across 4 datasets, GraphReach matches baselines in terms of accuracy while producing significantly more stable gene selections across seeds and remaining  robust to knowledge graph noise.

KEY STRENGTHS
- The problem is real and well-motivated. The stability effect is significant. The method is easy to implement and include to existing methods like GEARS and does not require extra data like IterPret.

WEAKNESSES
- The authors write "Our method chooses the
interventions that maximize the propagation of the supervision signal to the model, thereby
enhancing generalization" as if it was a causal link, but this claim is argued by analogy to previous work and never really tested.
- The reachability criterion is binary and discards both edge weights and gradient magnitude.
- Accuracy differences are negligible between baseline and other methods (Pearson reported x1000)

**Additional Comments:**

Some notations are used twice: M defines both the number of edges and the mapping function, while l defines both the loss and a graph layer.

The Conclusion is quite short. Could GraphReach generalize to different perturbation settings with different Knowledge Graphs? Can you detail your plan to leverage subset selection for foundation models?

**Audience:**

Yes

**Audience Explanation:**

Optimizing experimental design is a real-world problem and the paper findings would benefit the genetic perturbation practitionners.

**Broader Impact Concerns:**

I have no significant broader-impact concerns with this submission.

**Claims And Evidence:**

Yes

**Claims Explanation:**

I would say the claims are partially supported, and the one that are well supported are of interest.

- Efficiency wrt to active learning is very clear
- Stability is clearly measured through high gap in Jaccard. But could you demonstrate why it is biologically important to have a stable training set?
-Accuracy: no method appears distinguishable from the baseline, as Pearson scores are very close to each other. No statistical test reported.
- The claim that more reachability causes lower test error is not tested.

**Requested Changes:**

CRITICAL

- Evaluate the reachability to generalization mechanism directly. Do we observe a correlation between coverage and test error? Indeed, the current  accuracy evaluation shows very small differences compared the random baseline.
- Report accuracies with a table and confidence intervals - the x1000 plot is a bit misleading

STRENGTHEN
- Connect the gain in stability to a concrete downstream benefit.
- Comment on why does the reachability criterion use binary connectivity and not edge weights
- how do the results change with more than one GNN layer?

---

> ### Author Response · Authors · 2026-06-21
> **First answer to reviewer**
>
> We thank the reviewer for the careful points.
>
> - Stability is clearly measured through high gap in Jaccard. But could you demonstrate why it is biologically important to have a stable training set?
> - Connect the gain in stability to a concrete downstream benefit.
>
> We thank the reviewer for giving us a chance to clarify this.
> In general, we do not claim that stability is biologically important by itself but rather, the benefit is practical in the experimental-design setting. The selected perturbation set determines which genes will be assayed in the wet lab and therefore which costly training data will be generated. In this sense, the perturbation panel is an output of the method itself. If a model-based active-learning strategy produces substantially different panels under different random initializations, then the recommended experiment depends strongly on stochastic model artifacts rather than an inherent signal. Given the cost and duration of Perturb-seq experiments, this instability risks allocating wet-lab resources to experiments that are sensitive to arbitrary initialization choices.
> As a concrete downstream scenario, consider the design of a training set for graph-based perturbation prediction. With GraphReach, the selected perturbations are tied to the structure of the candidate perturbation space and to the graph neighborhoods used by the predictor. Therefore, it can provide a documented set for related studies with overlapping perturbation spaces, for example when extending to nearby pathways, related cell types or states, or improved graph-based perturbation models. In contrast, if the selected panel changes substantially across random initializations, it is not clear if it can be used as a starting point for related experimental designs.
> We will revise the manuscript to clarify that the downstream benefit of stability is reproducibility and reusability of the experimental design, not a direct biological claim about the selected genes.
>
>
> - Accuracy differences are negligible between baseline and other methods (Pearson reported x1000)
> - Accuracy: no method appears distinguishable from the baseline, as Pearson scores are very close to each other. No statistical test reported.
> - Report accuracies with a table and confidence intervals - the x1000 plot is a bit misleading
> - Indeed, the current accuracy evaluation shows very small differences compared the random baseline.
>
> We thank the reviewer for this comment. We agree that the original figure made the numerical differences harder to interpret as it was designed mainly to show the interaction between physical experimental time and accuracy. We added an explicit table reporting in the Appendix with the raw Pearson and MSE values with confidence intervals to compare the baseline with the GraphReach.
> Regarding the differences in accuracy, across the four datasets and two metrics, GraphReach improves over random selection in 6/8 evaluations, ties in 1/8, and is worse in 1/8. On average, the gain is $+4 \times 10^{-3}$ in Pearson and $-3 \times 10^{-3}$ in MSE.  These are of the same order as the ones reported in prior published work on active-learning for gene perturbation model training which we cite and base our evaluation and baselines on  (Huang et.al 2024). Specifically, Fig. 4 of Huang et. al reports gains of ≈ -3×10⁻³ MSE and +5×10⁻³ Pearson. Thus, the observed differences are consistent with the scale of improvements reported in the relevant active-learning literature for this problem.
> Finally, our claim is not that GraphReach produces large accuracy gains in isolation. Rather, it provides a promising tradeoff between random selection and active learning.

---

> ### Author Response · Authors · 2026-06-21
> **Second answer to reviewer**
>
> - The reachability criterion is binary and discards both edge weights and gradient magnitude.
> - Comment on why does the reachability criterion use binary connectivity and not edge weights
>
> We thank the reviewer for raising this point. We use binary reachability because GraphReach is intended to capture whether supervision can propagate to a node through the GEARS knowledge graph, not the exact strength of the eventual gradient update. In GEARS, the GO-based perturbation graph is constructed by computing Jaccard similarities between gene pathway sets and then keeping the top neighbors for each gene; our reachability criterion follows this sparsified graph structure and asks which embeddings can in principle receive supervision through message passing.
> We agree that edge weights and gradient magnitudes could provide a finer-grained criterion. However, gradient magnitudes are model- and training-dependent and are therefore not available in our one-shot pre-experiment setting, where perturbation labels have not yet been acquired. Our reachability criterion intentionally binarizes the edges because it is designed to capture whether supervision can reach an embedding through the receptive field, not the magnitude of the eventual update.  A weighted reachability objective is a natural extension, but it requires additional modeling choices, such as whether multi-hop edge strengths should be combined by products, sums, shortest weighted paths, random-walk probabilities, or another diffusion score. Different choices would encode different assumptions about how supervision attenuates through the graph. We therefore use binary reachability as the simplest model-output-free proxy aligned with the receptive-field argument, and will clarify this design choice in the revised manuscript while noting weighted reachability as future work.
>
>
> - The authors write "Our method chooses the interventions that maximize the propagation of the supervision signal to the model, thereby enhancing generalization" as if it was a causal link, but this claim is argued by analogy to previous work and never really tested.
> - The claim that more reachability causes lower test error is not tested.
> - Evaluate the reachability to generalization mechanism directly. Do we observe a correlation between coverage and test error?
>
> Thank you for pointing this out. We agree that the original wording could be read as implying a causal link between reachability and generalization. While this has been studied in the papers we refer to in section 3.2, we do not test it explicitly apart from testing the accuracy of the method.  Proposition 1 shows that, in the simplified GEARS graph-propagation module, supervision from selected perturbations can affect embeddings within their receptive fields. This motivates maximizing reachability, but it does not prove that higher reachability alone causes lower test error. Thus, in the revision, we changed the relevant statements in the Abstract, Introduction, Methodology, and Conclusion to describe reachability as an architecture-aligned proxy for potential supervision propagation, rather than as a proven causal mechanism for generalization.
>
>
> - How do the results change with more than one GNN layer?
>
> We did not run ablation studies on the GNN layers because that would compromise the accuracy of GEARS. In our experiments, we use the default GEARS configuration to ensure reproducibility and therefore set the layers accordingly.
>
> - Some notations are used twice: M defines both the number of edges and the mapping function, while l defines both the loss and a graph layer.
>
> We thank the reviewer for noticing this. We revised the notation to avoid overloading symbols.
>
> - The Conclusion is quite short. Could GraphReach generalize to different perturbation settings with different Knowledge Graphs? Can you detail your plan to leverage subset selection for foundation models?
>
> We thank the reviewer for this suggestion. We expanded the Conclusion to discuss how GraphReach could extend beyond the current GEARS setting. The method only requires a candidate perturbation set and a graph that defines relationships between perturbations, so it can in principle be applied with other biological knowledge graphs, such as pathway, regulatory, co-expression, or protein-interaction graphs, provided that the graph is relevant to the perturbation predictor. We also added a dedicated paragraph for the extension to single-cell foundation models.